# Effect of Topical Prostaglandin Analogue Therapy on Central Corneal Thickness: A Systematic Review

**DOI:** 10.3390/jcm12010044

**Published:** 2022-12-21

**Authors:** Jae-Yun Kim, Hyeon-Woo Yim

**Affiliations:** 1Graduate School, The Catholic University of Korea, Seoul 06591, Republic of Korea; 2Myung-Dong St. Mary’s Eye Center, Seoul 04534, Republic of Korea; 3Department of Preventive Medicine, College of Medicine, The Catholic University of Korea, Seoul 06591, Republic of Korea

**Keywords:** prostaglandin analogue, Latanoprost, Travoprost, Bimatoprost, Tafluprost, central corneal thickness, glaucoma, ocular hypertension

## Abstract

To investigate whether prostaglandin analogue (PGA) eyedrops have a significant effect on central corneal thickness (CCT), we conducted a systematic search of literature published from 2000 to 2021. Among the studies conducted on topical PGA therapy in open-angle glaucoma or ocular hypertension patients over 18 years old, prospective studies with CCT change as an outcome were included. A single-arm meta-analysis was conducted to assess the overall effect on CCT, and subgroup analysis according to exposure time of PGA eyedrops was also performed. We counted the number of articles that reported on severe events (CCT reduction of 25 μm or more) and obtained their proportion. The methodological quality was assessed by the McHarm tool. Twenty-two reports of prospective studies were selected. The results of the single-arm meta-analysis showed very high heterogeneity. Still, in subgroup analysis, when PGA was used for more than 6 months, heterogeneity was low, and a significant decrease in CCT was observed. Severe events were reported in two reports and occurred in 3.8% to 14.8% of participants. PGA eyedrop use may cause a clinically significant CCT decrease, requiring CCT follow-up.

## 1. Introduction

Prostaglandin (PG) is a lipid-signaling substance in the family of eicosanoids and is found in almost all tissues of humans and animals. Structural differences between PGs result in variable biological actions. Depending on the type of receptor to which even the same PG binds, the opposite action may appear in one tissue than in another. The PG F receptor in the eye is distributed in the ciliary muscle and trabecular meshwork. When it binds to PG F2a, it promotes the outflow of aqueous humor through the uveoscleral pathway, reducing intraocular pressure (IOP). Although the exact mechanism of this action is not well known, the PG F2a analogue (PGA), an agonist of the PG F receptor, upregulates matrix metalloproteinase (MMP) and degrades the extracellular matrix (ECM) of the ciliary muscle and sclera, reducing the resistance of uveoscleral outflow [1,2,3,4,5].

In 1996, Latanoprost, the first PGA eye drop, was approved for clinical use for glaucoma in the United States. After that, the FDA approved Travoprost, Bimatoprost, and Tafluprost as eye drops for glaucoma. PGA eye drops are currently the most widely used intraocular pressure-lowering agents. According to a meta-analysis, PGA eye drops showed a 24–33% reduction in IOP and, as a single agent, showed a stronger IOP lowering effect than other conventional glaucoma eye drops [6,7,8].

In glaucoma, central corneal thickness (CCT) has two important implications. The Ocular Hypertension Treatment Study (OHTS) suggested that CCT is a risk factor for glaucoma. In that study, for every 40 μm decrease in CCT, the incidence of glaucoma increased 1.71 times [9]. Second, since CCT affects the results of IOP measurement, a formula correcting for the effect of CCT is used. According to this, in CCT above 550 μm, the IOP increases by 2 mmHg for every 50 μm thickening [10]. Conversely, if the CCT is thinned, the measured IOP will be lower than the actual, underestimating the IOP. If this occurs while using glaucoma eye drops, the effect of the medication could be overestimated.

Lass et al. (2001) performed a randomized controlled trial (RCT) of the effects of PGA eyedrops on the cornea. In that study, CCT increased by 0.2 ± 3.1% in the group using Timolol eyedrops for one year, whereas CCT decreased by 1.1 ± 2.5% in the group using Latanoprost eye drops. Although the researchers concluded that there was no significant difference between the two groups because this reduction in CCT did not meet the pre-established criteria for significant change, this was the first report of a decrease in CCT with PGA eye drops [11]. Since then, several clinical trials and observational studies have reported that PGA eyedrops affect the cornea, and the CCT tends to decrease with use. Still, the significance of the change is inconsistent, and some studies have reported an increase. In 2016, Liu et al. reported CCT reduction as an adverse event in a systematic review (SR) of PGA eyedrops but cited only one RCT by Lass et al. [12]. Since then, CCT changes were not included as a safety issue in SRs for PGA eye drops until 2022.

If the CCT is significantly reduced with PGA eye drops, clinical consideration is necessary when prescribing it and during follow-up. Therefore, through a systematic review, the purpose of this study is to investigate whether the use of PGA eye drops has a significant effect on central corneal thickness change.

## 2. Materials and Methods

### 2.1. Search Strategy

We conducted a systematic search in MEDLINE, EMBASE, and Cochrane CENTRAL on literature published from 1 January 2000, to 14 June 2021. Although Latanoprost was first marketed in 1996, Lass et al. (2001) first raised the possibility that PGA eye drops could cause CCT reduction. Therefore, the literature search in this study started in 2000. In addition, reports found through Website search results (Google) and reports that met the criteria from the reference list of related SRs were included. Among the studies conducted on topical prostaglandin analogue therapy (Latanoprost, Travoprost, Bimatoprost, Tafloprost) in glaucoma (open-angle glaucoma (OAG), ocular hypertension (OHT)) those with patients over 18 years of age, and with central corneal thickness (CCT) change as an outcome were included. PGA eyedrops are used to lower intraocular pressure. Since CCT reduction corresponding to adverse effects is a safety issue, randomized controlled trials (RCTs), non-randomized studies (NRS: prospective observational study (cohort study), case-control, cross-sectional study), a systematic review (SR), a case report, and a case series were all included. The search terms and search formulas for each DB are attached in Appendix A (Table A1, Table A2 and Table A3).

### 2.2. Ethical Approval and Study Registration

This systematic review is subject to IRB review exemption, and the study protocol was exempted from participant consent from the Institutional Review Board of the Catholic University of Korea in May 2021. This systematic review protocol has been registered with PROSPERO, CRD42021258359.

### 2.3. Study Eligibility

For the primary outcome of CCT change after medication use, which is a safety-related subject, we conducted analysis and synthesis of only prospective studies (RCTs and prospective observational studies among the NRS), and the related case reports were reported separately.

Inclusion criteria were studies of the effect of PGA eyedrops on CCT for patients with OAG and OHT over 18 years of age. Studies were excluded for: (1) written in languages other than English, (2) not measuring CCT change, (3) non-prospective studies (case-control study, cross-sectional study, letter, review), (4) co-intervention (using only combination eyedrop of PGA and other ingredients, or undergoing surgical or laser treatment), (5) incomplete or unclear data, and (6) subjects with corneal abnormalities.

### 2.4. Types of Outcome Measure

Changes in CCT when using PGA eye drops were adverse events. Since CCT is a continuous variable, the mean, standard deviation (SD), and total number (N) were extracted, and the unit was micrometers. Finally, the mean difference was calculated and compared [13]. There are various methods to measure CCT, such as ultrasound pachymeter (USP), non-contact specular microscopy (NCSM), rotating Scheimpflug camera, optical low-coherence reflectometry (OLCR), and anterior OCT. USP is most commonly used as the gold standard. When measuring CCT with these five devices, intra-operator repeatability of each device in previous studies was evaluated by intraclass correlation coefficient (ICC), which was excellent at 0.9 or higher, indicating high repeatability [14,15]. Overall, strong agreement was shown between USP and each of the other four instruments with a Bland–Altman plot [15,16]. In the Bland–Altman plot between USP and NCSM, the 95% LoA range is relatively wide, so the agreement is weak. Therefore, measurement results are not interchangeable between devices, so CCT should not be measured with any other device during the study [14,17].

The concept of clinically significant CCT reduction is not yet established, but a change of 25 μm can be considered. With an average CCT of 550 μm, 25 μm corresponds to about 5% of the total thickness. In such a case, for every 25 μm decrease, the intraocular pressure is underestimated by about 1 mmHg [10]. Therefore, we consider a reduction of 25 μm or more as a clinically significant CCT decrease and defined it here as a severe event.

### 2.5. Study Selection

Two reviewers independently conducted the literature selection process for the identified records and the retrieved reports according to preset criteria. In the event of disagreement in the literature selection process, a third researcher was consulted, and a consensus was achieved.

### 2.6. Data Collection

Data collection was performed independently by two reviewers. From each document, authors, publication year, journal name, study design, study duration, total number of participants, inclusion criteria, clinical setting of the study, country, mean age, sex, number of intervention groups, primary outcome of the study, measurement method of CCT, time-point of measurement, scale of measurement, information of inspector, information of missing participants, key conclusions of authors, and funding source were extracted. The ingredient of eye drops used, exposure time to eyedrops, baseline IOP, number of participants allocated, baseline CCT, and change in CCT (mean, standard deviation) were collected from the PGA treatment group. For Meda et al. (2017), in which CCT variation was presented only as a graph rather than a numerical value, the graph was directly measured with WebPlotDigitizer version 4.4 (https://automeris.io/WebPlotDigitizer/, accessed on 8 June 2021.), and the value was estimated and extracted.

### 2.7. Quality Assessment

#### Assessment of Methodological Quality

As a study for safety evaluation, quality assessment was performed with the McMaster Quality Assessment Scale of Harms (McHarm) tool [18]. Among the 15 items of the McHarm tool, 9 suitable for this study were selected and applied. The nine selected McHarm elements are: whether (1) the harm was pre-defined, (2) severe events were defined precisely, (3) the study specified who collected the harm, (4) the study specified the training or background of who ascertained the harm, (5) the study specified the timing and frequency of collecting the harm, (6) the standard scale was used when collecting harm, (7) the number of participants that withdrew or were lost to follow-up was specified, (8) the total number of participants affected by harm was specified, and (9) the type of analyses undertaken for harms data was specified. Two reviewers evaluated all literature based on these selected questions.

### 2.8. Data Analysis and Synthesis

The extracted data were narratively synthesized and analyzed to assess the overall effect of PGA eyedrops on CCT change. Baseline CCT and follow-up CCT value, SD, and total N of the PGA treatment group underwent single-arm meta-analysis. A forest plot was drawn and visually shown to explore heterogeneity, and the i-square test was also performed. If the i-square value was greater than 25%, meta-analysis was performed using a random-effect model. To determine the effect of the duration of PGA eyedrop use on CCT change, a subgroup analysis was performed on studies that presented results within 4 months, 6 months, 12 months, and 24 months or more of eyedrop use. Since this study aimed to show the effect of PGA eye drops on CCT change, if each of the four kinds of PGA drugs was used as an independent treatment arm, the results of each PGA eyedrop were merged into one outcome and analyzed. Second, as a safety issue, we counted the reports that separately reported a reduction of 25 μm or more, which is considered here a clinically significant change in CCT. The proportion of subjects who showed a decrease of 25 μm or more among the total number of participants in the PGA treatment group from the counted reports was calculated. To evaluate publication bias for studies included in the meta-analysis, funnel plots were drawn. Statistical analysis was performed using Comprehensive Meta-Analysis (CMA) version 3 (Biostat, Inc., Englewood, NJ, USA).

## 3. Results

### 3.1. Search Results

Of 14,148 records searched via database and registers, 8913 were chosen. Among them, eligibility was assessed for 1018 reports, among which 20 reports were selected. Two reports were selected by assessing eligibility for ten reports out of eleven records retrieved from other sources. A systematic review was conducted on 22 prospective studies (Figure 1).

### 3.2. Study Characteristics

There were 22 prospective studies (*n* = 3880 participants). Among them, seven were RCTs, and one of the seven compared PGA with TML. Two of fifteen NRSs were secondary analyses of RCTs. These studies were conducted in 12 countries in Asia, Europe, and North and South America. Five studies were funded by pharmaceutical companies that manufacture PGA eye drops. In the 22 studies, the duration of medication use ranged from 1 month to 8 years. The components of PGA eyedrops used in the studies were LTP in 16 studies, TVP in 9, BMT in 7, and TAF in 1. However, three studies did not mention the eyedrop ingredients. For CCT measuring devices, USP was used in 15 studies, and non-contact type devices were used in six. Only Johny et al. (2020) did not mention the measurement instrument, or method, or the investigator who measured it [19] (Table 1).

### 3.3. McHarm Scale Assessment

The results of evaluating methodological quality with the McHarm tool are attached in Appendix A (Table A4). Of nine evaluation items, two reports scored seven points (highest), and five scored four (lowest). In only two studies, severe events precisely defined were reported [20,21].

### 3.4. Primary Outcome: Change in CCT

As a result of single-arm meta-analysis of CCT changes in the PGA eyedrop group in 22 reports (Figure 2), the overall CCT value showed a slight tendency to decrease. However, the i-square value was 95%, and the heterogeneity was quite high. We conducted subgroup analysis according to duration of PGA medication use to explore heterogeneity. As a result of subgroup analysis according to PGA eyedrop exposure period (Figure 3), i-square was 96% within 4 months. The heterogeneity was very high, so the synthesis result became meaningless. The heterogeneity was significantly lower in the results of subgroup analysis at 6 months, 12 months, and 24 months or more.

Of the 22 prospective studies, only 2 reported the number of participants with CCT reductions above a certain level [20,21]. At 12 months with PGA eye drops, about 5% of subjects showed a decrease of 25 μm or more in Panos et al. (2013). Approximately 5% of subjects showed a reduction of 30 μm or more, and 12% showed a 21–30 μm decrease in Schlote et al. (2009). To estimate a proportion of subjects whose CCT reduction was greater than 25 μm, the proportions were calculated by dividing the cases in which all 23 eyes that decreased by 21–30 μm in Schlote et al. (2009) were included or not. The rate of severe events was estimated at a maximum of 14.8 ± 1.6% when all 23 eyes were included and at a minimum of 5.1 ± 0.6% when all of them were excluded. (Table 2).

### 3.5. Publication Bias

The funnel plot of the 22 studies subjected to meta-analysis showed no publication bias (Figure A1).

## 4. Discussion

Long-term use of PGA eyedrops results in a decrease in CCT. This decrease was −5.30 ± 4.31 μm in the pooled estimate of prospective studies, which is small compared to the average CCT of 520–580 μm and seems to indicate low clinical significance. This suggests that long-term use of PGA eyedrops for glaucoma treatment is generally safe. However, since CCT reduction is a safety issue, general safety alone is not sufficient. Moreover, the heterogeneity of the synthesized result was very high. Among the expected causes of heterogeneity, clinical factors include PGA drug exposure time and type of PGA eyedrop, and methodological aspects include quality of literature. Therefore, we conducted a subgroup analysis of PGA medication exposure time. In the subgroup analysis, heterogeneity was significantly low from 6 months of use. That is, the CCT reduction effect of PGA within 4 months was unclear, but the significant decrease in CCT appeared after 6 months. According to these heterogeneity changes, CCT follow-up is necessary for glaucoma patients who have been using PGA eye drops for more than 6 months.

Although exposure time to the medication increases with time, CCT change also should be considered by the age of the subject. Brandt et al. (2008) and Viswanathan et al. (2013) analyzed the CCT change with aging. In a previous cross-sectional study of OHTS, CCT decreased by 0.63 μm per year [22], while that in a subsequent 3.8-year retrospective study, decreased by 0.74 ± 0.35μm per year [23]. In Viswanathan et al. (2013), CCT decreased by 0.45–0.55 μm per year [24]. Even considering such age-related CCT reduction, the decrease in CCT by PGA appears to be significant.

A clinically significant decrease of 25 μm or more, a severe event, would take 40 years or more to occur naturally, considering a reduction of 0.63 μm/year with aging. Clinically significant corneal thinning was not reported among 668 RCTs with PGA eyedrops, and severe events were reported by only 2 NRS among 22 published prospective studies on CCT changes of PGA eye drops. The proportion of subjects who showed a clinically significant decrease of 25 μm or more in these 2 studies ranged from 5.1 ± 0.6% to 14.8 ± 1.6% among PGA eyedrops users for 12 months. Considering that 1 mmHg of IOP is underestimated for every 25 μm decrease in CCT [10], 1 mmHg of IOP reduction may be overestimated in about 5 to 16% of PGA eyedrop users. The normal value of IOP is 10 to 21 mmHg, and in general, a 30% reduction is expected from PGA eyedrops, which is about 3 to 6 mmHg. Therefore, if 1 mmHg reduction is overestimated, 17–33% of the original treatment response is overestimated. However, since both studies were conducted on Caucasians in Switzerland, this 5% ratio cannot be directly applied to other population groups [20,21]. For some patients using PGA eyedrops, there is a possibility that the IOP-lowering effect may be significantly overestimated by CCT reduction, so regular CCT measurement is necessary during follow-up.

CCT reduction by exposure to PGA eyedrops appears to be reversible. In Meda et al. (2017), CCT increased after 6 weeks of discontinuation in subjects who had been using PGA eyedrops for more than 1 year and decreased again after 6 weeks of re-use. There were no baseline data before using PGA eyedrops, so we cannot know whether CCT had recovered to the level before the first use of the eyedrop. However, after 6 weeks of re-use, CCT decreased to the level of that achieved with use for more than 1 year [25]. Another retrospective study showed a decrease in CCT in patients with NTG who used PGA eyedrops for 5 years. Then, after 2 years of discontinuation, the CCT increased again, returning to the level before using PGA eyedrops [26].

There are three proposed mechanisms for CCT changes following PGA eyedrop instillation. First, PGA eyedrops increase the expression of IL (interleukin)-1 and IL-6, which are inflammatory cytokines on the ocular surface, and changes in these cytokines increase the production of MMP [27,28,29]. Increase in MMP-1 and MMP-9 and decrease in TIMP-1 occur on the ocular surface [30]. In Park et al. (2012) in which PGA eyedrops were administered to rabbits, MMP increased, and TIMP decreased in corneal stroma. This imbalance of MMP and TIMP reduced the extracellular matrix of the corneal stroma, including type 1 collagen, and caused a decrease in corneal thickness [27]. However, this reduction largely took place within 6 months of use of the medication, and there was little change after 2 years. Therefore, it is speculated that continued use of PGA eyedrops induces desensitization of PG F receptors and slows the increase in MMP [31]. Second, Liu et al. suggested that PGA eye drops, especially Latanoprost, could induce corneal fibroblast contraction in an in vitro study and affect corneal thickness reduction [32,33]. Finally, Shen et al. reported that Latanoprost might have a cytotoxic effect on corneal stromal cells, inducing apoptosis and thus reducing corneal thickness [32,34]. The proposed mechanism for CCT increase during PGA eyedrop use is as follows. Latanoprost instillation increases the intracellular calcium ion concentration and activates the Protein Kinase C pathway. Then, the cell shape changes to round, which may change the corneal cells and increase the CCT [35].

In a cross-sectional study using Casia OCT by Amano et al. in 2019, when PGA eyedrops were used in patients with open-angle glaucoma without corneal disease or surgery, there were changes in the biomechanical factors of the cornea, including CCT. Still, there was no change in the shape itself [36]. However, due to the effects of PGA eyedrops on the cornea other than CCT reduction, Amano et al. in 2008 reported a case in which the disease progressed after PGA eyedrop treatment in a keratoconus patient [37]. In 2009, Nowroozzadeh et al. said that PGA eyedrop treatment adversely affected the progression of myopic regression in patients who underwent corneal refractive surgery [38]. In 2020, Rodrigo-Rey, S. et al. reported a case in which, with open-angle glaucoma without corneal disease such as keratoconus or previous corneal surgery, visual acuity decreased after using PGA eyedrops for 1 year due to mild corneal ectasia, but visual acuity and cornea returned to normal after changing the medication [39]. This means that long-term PGA use may, although rarely, change the shape of the cornea, in addition to changes in corneal biomechanical factors including CCT. In long-term use of PGA eye drops, regular monitoring of corneal topography is necessary in addition to CCT measurements Moreover, when selecting PGA eye drops for patients with underlying diseases such as pathologic corneal thinning involving keratoconus or after refractive surgery, it is necessary to consider the possibility of such changes and provide sufficient prior explanation to the patient.

The limitations of this study are that, first, the analysis was not able to be conducted on comparative studies between the two groups. In the studies for PGA eye drops, TML eyedrops were usually used as a control group, which is appropriate to evaluate the effect of lowering IOP. However, in studies for CCT change, since TML is also known to affect CCT change, it is not suitable as a control. Therefore, this study performed a single-arm meta-analysis only on the PGA treatment group. Second, there is a lack of studies on CCT changes with PGA eyedrops. Two studies which reported severe events were NRS, and no RCT reported severe events. In future prospective studies on PGA eye drops, it is necessary to consider CCT changes in advance. Third, in this SR, analysis according to type of PGA medication could not be performed. Finally, although this review’s pooled estimate of CCT change was small compared to the average corneal thickness, only 2 NRS studies reported the proportion of subjects who showed a clinically significant decrease. Therefore, it is necessary to conduct a large-scale RCT for CCT change following PGA eyedrop use over 2 years. In that RCT, it will be essential to identify the percentage of subjects showing a clinically significant decrease in CCT after classifying the subjects according to whether CCT has decreased and the amount of decline.

## Figures and Tables

**Figure 1 jcm-12-00044-f001:**
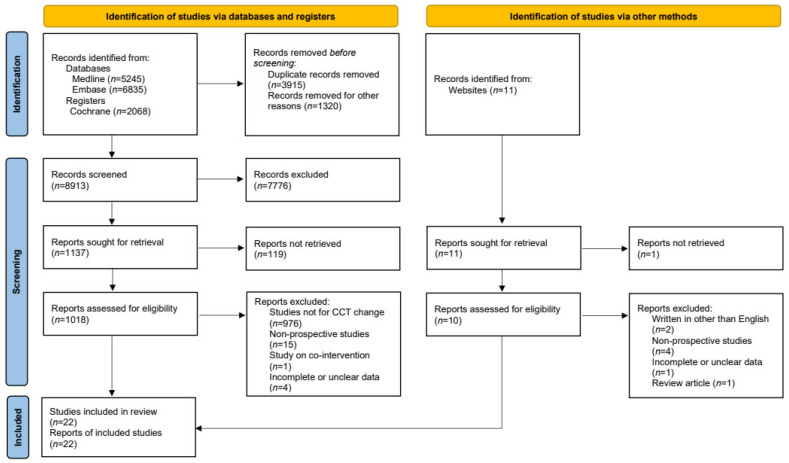
PRISMA 2020 flow diagram of the literature search process for identifying studies. CCT—central corneal thickness.

**Figure 2 jcm-12-00044-f002:**
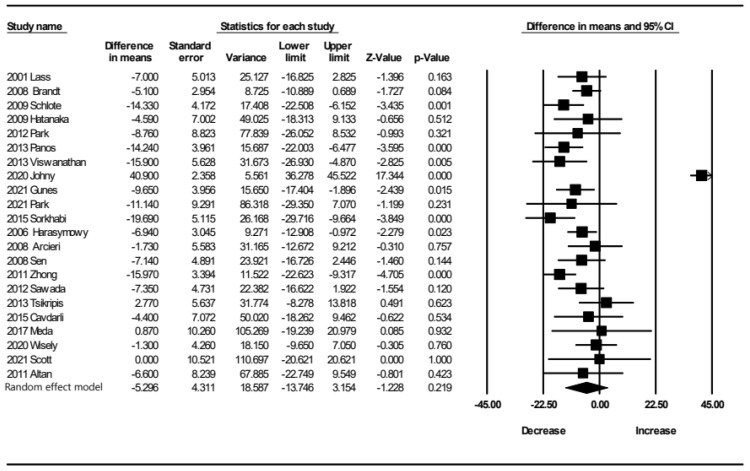
Single-arm meta-analysis of CCT change after topical prostaglandin analogue treatment.

**Figure 3 jcm-12-00044-f003:**
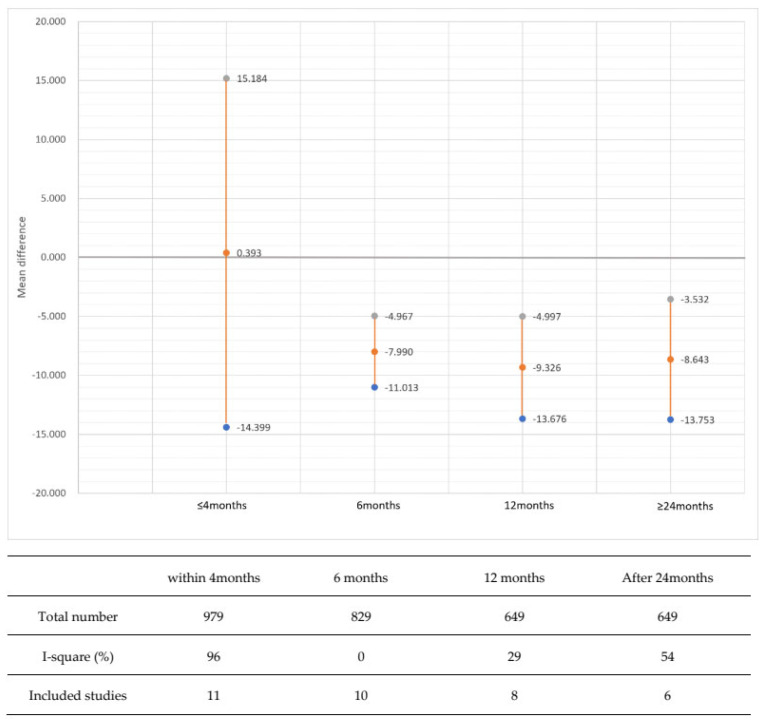
Subgroup analysis according to duration of topical prostaglandin analogue treatment.

**Table 1 jcm-12-00044-t001:** Characteristics of the selected 22 prospective studies.

Author	Publication Year	Study Design	Country	Total *n* of Participants	Age	Sex	Diagnosis	Baseline IOP (mmHg)	Ingredients of Eyedrops Used	Duration of Medication	The Instrument for Measuring CCT	Funding Source
Lass	2001	RCT	US	369	61 ± 12	M: 183 F: 186	OAG, OHT	19.6 ± 3.5	LTP	12 mon	USP	Pharmacia corp.
Brandt	2008	Secondary analysis of RCT	US	1191	60.4 ± 9.1	M: 506 F: 685	OHT	25.0 ± 2.6	PGA	5.0 ± 2.7 yrs	USP	NEI
Schlote	2009	Prospective NRS	Switzerland	94	66.24 ± 13.73	M: 45 F: 49	Glaucoma	NM	TVP	12 mon	OLCR	none
Hatanaka	2009	RCT	Brazil	73	68.5 ± 9.2	NM	OAG	NM	LTP, TVP, BMT	8 wks	USP	none
Park	2012	retrospective cohort study	Republic of Korea	44	63.74 ± 9.8	M: 29 F: 31	Glaucoma	18.63 ± 4.91	LTP, TVP, BMT	2.47 ± 0.66 yrs	USP	none
Panos	2013	Prospective NRS	Switzerland	90	67.8 ± 15.3	M: 49 F: 41	POAG	22.58 ± 2.94	TAF	12 mon	NCSM	none
Viswanathan	2013	Prospective NRS	Austrailia	287	68 ± 10.14	M: 138 F: 163	POAG, NTG	21.68 ± 4.64	PGA	6.92 ± 1.67 yrs	USP	none
Johny	2020	Prospective NRS	India	80	46.6 ± 8.88	NM	POAG, OHT	25.57 ± 1.28	LTP	4 mon	NM	none
Gunes	2021	Prospective NRS	Turkey	222	54.1 ± 4.30	M: 30 F: 34	POAG	24.6 ± 2.2	LTP	36 mon	OCT	none
Park	2021	Prospective NRS	Republic of Korea	65	51.35 ± 13.27	NM	POAG	17.9 ± 3.8	PGA	12 mon	NCSM	Korea University
Sorkhabi	2015	prospective NRS	Iran	90	64.5 ± 10.86	M: 46 F: 16	OAG	NM	LTP	12 mon	USP	NM
Harasymowycz	2006	Prospective NRS	Canada	379	67.0 ± 11.4	M: 157 F: 222	OAG, OHT	24.66 ± 5.16	TVP	6 wks	USP	Alcon Canada
Arcieri	2008	RCT (cross-over)	Brazil	34	57.8 ± 9.6	M: 14 F: 20	POAG, OHT	22.6 ± 3.3	BMT, LTP, TVP	4 wks for each medication	USP	Alcon, Norvatis, Pfizer
Sen	2008	RCT	Turkey	94	56.05 ± 11.76	M: 37 F: 57	OAG, PXG, NTG, OHT	23.38 ± 4.11	LTP, BMT	24 mon	USP	NM
Zhong	2011	RCT	China	69	51.04 ± 12.31	M: 35 F: 34	OAG, NTG, OHT	23.23 ± 3.44	LTP, TVP, BMT	17.56 ± 15.68 mon	USP	none
Sawada	2012	RCT (cross-over)	Japan	42	53.2 ± 11.8	M: 23 F: 19	OAG	14.4 ± 2.6	LTP, TVP	12 wks for each medication	USP	none
Tsikripis	2013	RCT	Greece	108	61.82 ± 10	NM	OAG	17.59 ± 3.91	LTP	more than 3 yrs	USP	none
Cavdarli	2015	Prospective NRS	Turkey	24	58.5 ± 12.6	NM	POAG, PXG	25.0 ± 4.2	LTP	6 mon	USP	NM
Meda	2017	Prospective NRS	Canada	35	69.0 ± 1.6	M: 16 F: 19	POAG	22.23 ± 0.95	BMT, LTP, TVP	3.99 ± 0.28 yrs	USP	Allergan and Alcon
Wisely	2020	Post-hoc analysis of RCT	USA	415	63 ± 11.2	M: 156 F: 259	OAG, OHT	NM	LTP	3 mon	USP	Aerie Pharmaceuticals
Scott	2021	Prospective NRS	USA	21	60 (36~79)	M: 12 F: 21	OAG	21.7 ± 6.2	LTP, BMT, TVP	4 mon	RSC	Columbus Foundation
Altan	2011	Prospective NRS	Turkey	54	59.2 ± 12.6	M: 21 F: 33	OAG, OHT	21.4 ± 7.2	LTP	6 mon	Corneal Topography	NM

*n* (number), yrs (years), mon (months), wks (weeks), IOP (intraocular pressure), RCT (randomized controlled trial), NRS (non-randomized study), NM (not mentioned), OAG (open-angle glaucoma), OHT (ocular hypertension), POAG (primary open-angle glaucoma), NTG (normal tension glaucoma), PXG (pseudoexfoliative glaucoma), PGA (prostaglandin analogue), LTP (Latanoprost), TVP (Travoprost), BMT (Bimatoprost), TAF (Talfuprost), USP (ultrasonic pachymetry), OCT (optical coherence tomography), OLCR (optical low coherence reflectometry), NCSM (non-contact specular microscopy), RSC (rotating Scheimfplug camera).

**Table 2 jcm-12-00044-t002:** Frequency of severe events after 12 months of PGA medication in two studies reporting severe events.

	2009 Schlote (136 Eyes)	2013 Panos (100 Eyes)
Decrease in CCT ^1^		
21 to ≤30 µm	23 eyes (16.9%)	12 eyes (12.0%)
>30 µm	7 eyes (5.1%)	2 eyes (2.0%)
>25 µm *	7–30 eyes (5.1–22.0%) ^2^	5 eyes (5.0%)

* Definition of severe events in this SR. ^1^ CCT (central corneal thickness). ^2^ Schlote et al. (2009) [20] did not separately report a reduction of more than 25 μm. To estimate the proportion of subjects whose CCT reduction was greater than 25 μm, the proportions were calculated by dividing the cases in which all 23 eyes that decreased by 21–30 μm were included or not.

## Data Availability

Not applicable.

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
