# Peer review of "Effect of Topical Prostaglandin Analogue Therapy on Central Corneal Thickness: A Systematic Review"

_jcm, 2022, doi:10.3390/jcm12010044_

Round 1
Reviewer 1 Report
In this review the authors discuss the effect of PGAs on the central corneal thickness. Below are my remarks:
Lines 267-272: The authors state:” In a previous cross-sectional study of OHTS, CCT decreased by 0.63μm per year, while that in a subsequent 3.8-year retrospective study, decreased by 0.74±0.35μm per year.[27]” In which study did the CCY decreased by 0.633m per year? Viswanathan et al. (2013) (ref 28) compared a group of patients receiving PGA or PGA/b-blockers with a control group of glaucoma suspects who did not receive any medication. The authors need to clarify to which studies they refer and discuss in further detail ref 28
This review contains a lit of information on the effect of PGAs on CCT, it is well presented and adequately discussed. The methodology is clearly written
Reviewer 2 Report
Kim & Yim provide a systematic review of an important topic -- CCT changes with topical prostaglandin analogue therapy. The report is well-designed, well-written, and adequately describes the current body of work in this area.
I have no editorial comments but want to encourage the authors to:
1. Place the [ ] for references inside the sentences (e.g., for line 36, should read "....outflow [1-5].")
2. Why separate the Google search (in line 74) from the rest?
Interesting paper. Thanks for allowing me to review it.
